# Effectiveness of Psychobiotics in the Treatment of Psychiatric and Cognitive Disorders: A Systematic Review of Randomized Clinical Trials

**DOI:** 10.3390/nu16091352

**Published:** 2024-04-30

**Authors:** Freiser Eceomo Cruz Mosquera, Santiago Lizcano Martinez, Yamil Liscano

**Affiliations:** 1Grupo de Investigación en Salud Integral (GISI), Departamento Facultad de Salud, Universidad Santiago de Cali, Cali 760035, Colombia; 2Área Servicio de Alimentación, Área Nutrición Clínica Hospitalización UCI Urgencias Y Equipo de Soporte nutricional, Clínica Nuestra, Cali 760041, Colombia; lizmasan15@hotmail.com

**Keywords:** psychobiotics, gut–brain axis, mental health disorders, neurotransmitters, inflammation, systematic review

## Abstract

In this study, a systematic review of randomized clinical trials conducted from January 2000 to December 2023 was performed to examine the efficacy of psychobiotics—probiotics beneficial to mental health via the gut–brain axis—in adults with psychiatric and cognitive disorders. Out of the 51 studies involving 3353 patients where half received psychobiotics, there was a notably high measurement of effectiveness specifically in the treatment of depression symptoms. Most participants were older and female, with treatments commonly utilizing strains of Lactobacillus and Bifidobacteria over periods ranging from 4 to 24 weeks. Although there was a general agreement on the effectiveness of psychobiotics, the variability in treatment approaches and clinical presentations limits the comparability and generalization of the findings. This underscores the need for more personalized treatment optimization and a deeper investigation into the mechanisms through which psychobiotics act. The research corroborates the therapeutic potential of psychobiotics and represents progress in the management of psychiatric and cognitive disorders.

## 1. Introduction

The global increase in psychiatric and cognitive disorders has significantly raised concerns within the medical community, impacting both students and the general population profoundly [1,2,3]. The challenges posed by academic demands, family pressures, and the competitive nature of the medical field contribute to considerable psychological stress, potentially leading to severe disorders like depression and anxiety [2]. In 2019, the World Health Organization (WHO) highlighted that approximately one in eight individuals, or around 970 million people worldwide, were afflicted with a mental disorder, with those related to anxiety and depression being the most widespread [4].

In addition to these problems, schizophrenia has also been generating significant concern. This mental health condition, characterized by distortions in thinking, perception, emotions, language, sense of self, and behavior, adds another layer of complexity to the already daunting challenge of addressing mental health globally [5]. The incidence of schizophrenia, while not as high as that of anxiety and depression, still presents a substantial public health issue, requiring targeted interventions and comprehensive support systems to manage its symptoms and improve the quality of life for those affected [6,7]. Existing approaches to treating schizophrenia, which encompass both medication-based and therapeutic strategies, successfully alleviate symptoms. However, there is a recognized necessity for more specific measures aimed at mitigating negative symptoms and enhancing cognitive functions [8]. The confluence of these mental health disorders underscores the need for a more integrated approach to mental healthcare, prioritizing early detection, personalized treatment plans, and the de-stigmatization of mental health conditions to foster a more understanding and supportive society [9,10,11].

In this context, a new focus of attention emerges: psychobiotics, a subset of probiotics that promises to benefit mental health by modulating the intestinal microbiota [12]. These beneficial microorganisms have caught the interest of the scientific community due to their potential to alleviate symptoms associated with depression and anxiety, diminish the body’s response to stress, and enhance cognitive functions such as memory [12,13]. Unlike traditional approaches, psychobiotics represent an approach that aligns with the current preference for more natural treatments and with fewer side effects [14,15].

Probiotics, defined as live microorganisms that, when administered in adequate amounts, confer a health benefit on the host, have been used for decades to improve gastrointestinal health [16,17]. Psychobiotics, on the other hand, specifically focus on mental health, acting on the intestinal microbiota to influence neurological and cognitive processes [12,18].

The interaction between psychobiotics and the gut–brain axis is a burgeoning field of research with significant implications for mental health and general wellness. Psychobiotics, encompassing both prebiotics and probiotics, play a vital role in the production of neurotransmitters such as dopamine, norepinephrine, GABA, serotonin, and acetylcholine, all of which are crucial for the brain and central nervous system’s (CNS) optimal functionality. These microorganisms, along with their metabolites—especially short-chain fatty acids (SCFAs)—are key to colon health, the regulation of inflammation, and metabolic processes, affecting the blood–brain barrier’s permeability and tryptophan’s metabolism into serotonin [19].

The gut microbiota is integral to intestinal functionality, digestion, and nutrient absorption, producing SCFAs like butyric acid, which is essential for colon cell growth. It plays a pivotal role in detoxifying harmful substances, establishing an intestinal barrier, and modulating the immune system. Yet, disturbances in the gut microbiota balance can trigger various health issues, including immune system overactivation and diseases [20].

Studies have shed light on how the gut microbiota influences brain functionality, linking it to conditions such as schizophrenia via interactions with the immune system and neurotransmitter effects. Factors like inflammation, stress, and circadian rhythm disruptions can significantly alter the gut microbiota, diminishing beneficial bacteria and heightening susceptibility to disease. Furthermore, environmental influences, dietary shifts, and exposure to pollutants considerably affect gut microbiota composition, connecting these changes to the onset of psychiatric conditions [21].

Psychobiotics also bolster hormone production, including cholecystokinin (CCK), peptide tyrosine tyrosine (PYY), and glucagon-like peptide-1 (GLP-1), and influence neurotransmitter synthesis through the interaction of the enteric nervous system (ENS) with the CNS via the vagus nerve [22,23,24]. The corticotropin-releasing hormone (CRH), which is critical for gastrointestinal functionality and stress-related disturbances, along with certain lactic acid bacteria’s ability to produce neurotransmitters like nitric oxide (NO), highlight the complex relationship between the nervous system and gut microbiota [24,25].

The action mechanisms of psychobiotics encompass safeguarding intestinal barrier integrity, activating the immune system, altering the host’s microbiota, modulating metabolic reactions, and affecting the CNS [26]. This includes improving tight junction proteins for intestinal health, fostering immune system development, and generating compounds like conjugated linoleic acid (CLA) that regulate tight junction proteins and antioxidant enzymes, thus mitigating oxidative stress and inflammation [21,27,28].

Understanding this intricate connection between psychobiotics and the gut–brain axis illuminates potential probiotic-based therapeutic approaches for neuropsychiatric disorders. It underscores the importance of a healthy gut microbiota for mental health maintenance and overall well-being, suggesting a holistic approach to health that integrates gut and brain health [22,29,30].

However, despite these promising findings, there is a significant deficit in the systematic and detailed understanding of the efficacy of psychobiotics. There is a critical need for additional research to determine optimal doses, the most effective strains of psychobiotics, and their exact mechanisms of action.

This manuscript aims to address these gaps in research, providing a systematic review of existing randomized clinical trials on the effectiveness of psychobiotics in the treatment of psychiatric and cognitive disorders. By doing so, we seek not only to shed light on the current evidence but also to highlight the importance of this emerging area of study and its potential impact on global mental health.

## 2. Materials and Methods

### 2.1. Study Protocol

This systematic review was conducted in accordance with the guidelines of the Cochrane Collaboration Handbook and reported considering the recommendations for systematic reviews and meta-analyses of the PRISMA statement [31]. This research was formulated considering the PICO strategy (Population, Intervention, Comparison, Outcomes) [1].

### 2.2. Research Question

Is the administration of psychobiotics (I) compared to placebo (C) in adults with psychiatric and cognitive disorders (P) effective in reducing symptomatology or improving quality of life (O)?

### 2.3. Eligibility Criteria

#### 2.3.1. Inclusion Criteria

The inclusion criteria were:Randomized controlled trials published between January 2000 and December 2023.Studies published in any language.Human studies on individuals over 18 years evaluating the effects of psychobiotics (probiotics, prebiotics, and symbiotics) administered in any form (capsule, fermented food) alone or combined, as an intervention in psychiatric and cognitive disorders.Studies with the primary outcome of quantitative data on improvement in symptoms, quality of life, or adverse effects in patients presenting with a primary or comorbid psychiatric or cognitive disorder diagnosed before entering the study.

#### 2.3.2. Exclusion Criteria

Articles in pre-print mode or letters to the editor.Studies not available in an accessible format.Studies that do not specify the strain of psychobiotic used.Studies investigating the use of probiotics as part of a comprehensive approach, where it is difficult to separate and specifically understand how much psychobiotics contribute compared to other interventions.Studies containing findings that have already been published as part of a previous publication (post hoc analyses that revisit preliminary findings).Studies of structural and neurodegenerative disorders, including multiple sclerosis, Alzheimer’s, Huntington’s, ALS, and Parkinson’s.Studies with pregnant patients.

### 2.4. Data Sources and Search Strategy

This search was conducted in databases: Pubmed, Cochrane Clinical trial, SCOPUS, Science Direct, Biomed Central (BMC), Web of Science, Springer, and the Virtual Health Library (VHL). No language filters were applied, and the date range was set between 2000 and 2023. The search strategy was designed and executed from November 2023 to January 2024 by two researchers independently (F.E.C.M and Y.L.), using the following keywords: prebiotics, probiotics, psychobiotics, psychiatric probiotics, bifidobacteria, lactobacilli, mental health, mental well-being, depression, anxiety, mental illness, psychiatric disorder, psychiatric illness, clinical trial, and randomized controlled trial. Terms were combined using the Boolean operators AND and OR (see search details in Appendix A). References of relevant articles were reviewed, and additional web searches were conducted to identify studies not evident through initial tracking. When necessary to confirm the clinical trial or expand the information of a study, access to ClinicalTrails.gov (https://clinicaltrials.gov/ accessed on 18 November 2023) was sought, if applicable. Data were stored using Zotero version 6.0.

### 2.5. Selection and Data Extraction

The selection of potentially eligible studies was carried out independently by the two researchers, initially examining the title and abstract, and subsequently the full text. Studies with doubts regarding their relevance were discussed thoroughly, and the decision to include them in this review was made by consensus. A reviewer (F.E.C.M) extracted information from the primary studies considering details of the clinical trial (first author, year of publication), characteristics of the participants (number of subjects assigned to each group, diagnosis, average age plus standard deviation, percentage of female participants), design of the trial (psychobiotic used, presentation, dosage, weeks of treatment), and results (primary outcome, psychometric scale used, safety measured through the frequency of adverse events). Subsequently, a second reviewer (Y.L.) verified the integrity and accuracy of the recorded information.

### 2.6. Risk of Bias Assessment

The risk of bias assessment for the primary studies was conducted independently using a standardized instrument that considers the essential elements of a clinical trial’s design. For the assessment, data were entered into Review Manager version 5.4^®^ (RevMan, accessed on 23 November 2023). The criteria considered were (a) generation of random sequence, (b) concealment of allocation, (c) blinding of participants and personnel, (d) masking of outcome assessment, (e) incomplete outcome data, and (f) selective reporting [32]. For each domain assessed, it was considered that the RCTs presented a low or high risk of bias, according to whether or not they complied with predetermined guidelines. Discrepancies in the risk of bias assessment were resolved through discussions between the reviewers until an agreement was reached. To confirm the consistency of the evaluation process, a subset of the studies was reassessed, and the Cohen’s kappa coefficient was computed to quantify the agreement [33].

### 2.7. Ethical Considerations

During the development of this study, no interventions were made in the demographic and physiological variables of the participants. Therefore, this research work is considered minimal risk according to Resolution No. 8430 of 1993 of Colombian legislation and the Declaration of Helsinki.

## 3. Results

### 3.1. Characteristics of the Included Studies

After searching the aforementioned databases, a total of 5360 potentially relevant articles were found. Following the removal of 239 texts due to duplication, 4449 articles were screened by title and an additional 583 were eliminated after abstract review. Out of the 89 research studies, 37 were excluded for the following reasons: (a) they were not randomized controlled trials (n = 4), (b) they involved underage patient populations (n = 2), (c) even though they measured changes in the intestinal microbiota or biomarkers, they did not evaluate outcomes previously established in the review (n = 24), (d) they were not available in accessible formats (n = 1), (e) the population consisted of pregnant women (n = 4), or (f) they evaluated patients with neurodegenerative or structural diseases such as Huntington’s or Parkinson’s (n = 2). A Cohen’s kappa of 0.96 for records excluded by title and abstract suggested a high level of agreement, indicating that the reviewers were almost in perfect accord about which records to exclude based on the initial screening. Similarly, a kappa of 0.87 for reports excluded after full-text assessment signified a substantial agreement, albeit slightly lower than for the initial screening, which was expected as decisions at this stage are often more complex and subjective. These results overall reflected a very high degree of reliability and consistency between the evaluators in the study selection process. Ultimately, 51 articles were included in this review. The details of the study selection are found in the PRISMA flowchart (see Figure 1).

### 3.2. Findings in the Studies

The characteristics of the articles included in this review are detailed in Table 1. All the studies were randomized controlled trials with a total population of 3353 patients, averaging 67 ± 12.6 participants per study. More than half of the global population studied (n = 1871) received treatment with probiotics, prebiotics, or symbiotics. The clinical condition most frequently evaluated for the effect of psychobiotics was depression (52.46%) [32,33,34,35,36,37,38,39,40,41,42,43,44,45,46,47,48,49,50,51,52,53,54,55], followed by cognitive impairment (10%) [56,57,58,59,60], schizophrenia (10%) [61,62,63,64,65], and bipolar I affective disorder (6%) [66,67,68,69].

The average age of participants in the RCTs ranged from 22.2 to 72.2 years, and was typically older for patients involved in cognitive impairment research. Additionally, in more than half of the studies, the female population constituted the majority of the sample (n = 40).

Regarding therapeutic measures, 44 studies used probiotics, 3 implemented prebiotics, and 5 considered a mix of both (symbiotics). The most used probiotics belonged to the Lactobacillus and Bifidobacteria families. However, the product supplied usually consisted of multiple species. Regarding prebiotics, authors used substances like inulin, galacto-oligosaccharide, 4G-beta-D-galactosylsucrose, or provided diets rich in prebiotics.

The treatment regimen with psychobiotics in the selected RCTs varied between 4 and 24 weeks, with the majority having a duration of 4–12 weeks. Regarding the scales or questionnaires used to measure patient symptom improvement, for depression, the 24-item Hamilton Rating Scale for Depression, the Beck Depression Inventory II, the Montgomery–Åsberg Depression Rating Scale, and the Depression, Anxiety and Stress Scale 42 were implemented. For schizophrenia, the Positive and Negative Syndrome Scale and the Brief Psychiatric Rating Scale were used. Anxiety improvement was assessed with changes in the Hamilton Rating Scale for Anxiety and the Beck Anxiety Inventory. Regarding bipolar affective disorder, symptom evolution was measured with multiple instruments, the most specific being the Young Mania Rating Scale (Table 1).

### 3.3. Safety of Psychobiotics

In the analysis of adverse events reported in the studied population (see Table 2) through various trials on psychobiotics, a general pattern of good tolerance towards these products is observed. From the studies reviewed, ranging from Dickerson F. et al. [67] in 2014 to Freijy T. et al. [81] in 2023, zero to 75 adverse events were reported in patients treated with psychobiotics. Common adverse events included gastrointestinal symptoms such as constipation, nausea, abdominal pain, gastrointestinal discomfort, flatulence, and changes in appetite, in addition to other effects like headache, anxiety, and sleep disturbances.

However, the majority of these events were classified as mild, and very few studies reported serious adverse events that led to a participant withdrawing from the study. For example, one serious event was reported in the probiotic group in the study by Dickerson F. et al. [67] in 2014, but it was not directly related to the product. In the case of Hwang Y. et al. [61] in 2019, a serious adverse event reported in the probiotic group required the participant to withdraw for treatment.

Studies such as those by Pinto et al. [74] in 2017, Rudzki et al. [77] in 2018, and Vaghef E. et al. [44,45] in 2021 and 2023 indicate that the adverse events presented were resolved in less than two weeks or were not serious. Moreover, several studies, including those by Majeed M et al. [48] in 2018, Kobayashi Y. et al. [62] in 2019, and Gualtieri et al. [72] in 2020, reported no adverse events related to the intake of probiotics.

This data set reinforces the safety of psychobiotics as treatment, highlighting that while adverse events can occur, they are usually of a mild and manageable nature. The scarcity of serious adverse events and the good tolerance observed in the studies underscore the potential of psychobiotics as a safe therapeutic option for treating various psychiatric and cognitive conditions.

### 3.4. Risk of Bias Assessment

Based on the risk of bias graph generated using RevMan 5.4^®^ (accessed on 23 November 2023), the assessment of bias risk for the included studies can be summarized as follows:

The evaluation of bias risk in these studies was conducted using various criteria, as depicted in Figure 2. For the aspect of random sequence generation (selection bias), the majority of studies were deemed to have a low risk, indicating that the randomization procedures were well executed and thoroughly documented. Similarly, allocation concealment (selection bias) was predominantly assessed as low risk, suggesting that the allocation process was adequately concealed to minimize selection bias.

Regarding the blinding of participants and personnel (performance bias), the risk was mostly low, implying effective blinding of participants and study personnel to treatment allocations, thereby reducing the likelihood of performance bias. The blinding of outcome assessment (detection bias) also primarily presented a low risk, indicating that outcome assessors were probably unaware of the intervention groups, which minimizes detection bias.

However, the domain concerning incomplete outcome data (attrition bias) exhibited a range of low, unclear, and high risks across the studies. A low risk in this area indicates transparent reporting of participant dropouts and proper management of missing data. Conversely, studies with an unclear risk lacked detailed reporting on attrition and how missing data were handled, while a high risk pointed to a transparency deficit that could affect the validity of study outcomes.

The issue of selective reporting (reporting bias) also showed variability, with most studies being classified as low risk. This classification suggests that it is likely all predefined outcomes were reported and the study protocol was registered. However, some studies had an unclear risk due to inadequate information regarding the reporting of all expected outcomes.

In summary, the overall risk of bias for the studies was primarily low across most evaluated domains, reflecting a high methodological quality. Nonetheless, variability was observed, particularly concerning incomplete outcome data and selective reporting, warranting cautious interpretation of the findings from these studies.

To enhance the robustness of future research, it is crucial to explicitly describe randomization and concealment techniques; enforce and document blinding for all participants and assessors; rigorously report and manage incomplete data; commit to registered protocols to prevent selective reporting of outcomes; and consider all potential biases. Following these guidelines will strengthen the reliability and applicability of research conclusions.

## 4. Discussion

### 4.1. Main Findings

The systematic review conducted aimed to consolidate and evaluate the evidence from RCTs regarding the effectiveness of psychobiotics in treating psychiatric and cognitive disorders. Despite the growing interest and preliminary positive findings in this field, there is a noticeable lack of comprehensive understanding regarding their true efficacy, optimal dosing, most effective strains, and precise mechanisms of action.

This study screened a vast number of articles, with 51 studies making it to the final review, covering a total of 3353 patients. The predominant clinical condition studied in relation to psychobiotics was depression, followed by cognitive impairment, schizophrenia, and bipolar I disorder. The studies varied in treatment durations, ranging from 4 to 24 weeks, with the most common duration being 4 to 12 weeks. The patient demographic included an average age range from 22 to over 70 years, with a notable majority of female participants across more than half of the studies.

Most studies used probiotics from the Lactobacillus and Bifidobacteria families, and several studies used prebiotics or a combination of both. Various scales and questionnaires were employed to measure symptom improvement, with the Hamilton Rating Scale for Depression and the Beck Depression Inventory II being common for depression, and other specific scales used for different disorders.

The risk of bias assessment highlighted that most studies had a low risk across several domains, including random sequence generation and blinding of outcome assessment, suggesting methodological rigor. However, there were inconsistencies in reporting incomplete data and potential selective reporting, indicating a need for more thorough documentation and adherence to registered protocols in future research.

### 4.2. Variety in Psychiatric Disorders and Their Treatments

This systematic review emphasizes the importance of discussing the heterogeneity of the psychiatric disorders addressed and the diversity of available treatments to justify the inclusion of psychobiotics as a valid and potentially revolutionary therapeutic strategy in psychiatry. The role of psychobiotics should be explored across a broad spectrum of psychiatric conditions to realize their extensive therapeutic potential and ensure their widespread applicability. This not only involves identifying the most common strains and their action mechanisms for effective comparison but also standardizing strains and dosages to maximize research reproducibility and efficacy [85,86]. Through their influence on gut microbiota and the brain–gut axis, psychobiotics offer a multifaceted approach that could be particularly beneficial in managing these disorders [25]. Their ability to modulate neurotransmitters such as tryptophan and serotonin and their anti-inflammatory effects can substantially improve psychiatric symptoms related to cerebral inflammatory states [52]. Subsequently, it can be observed that the effects of psychobiotics on various disorders exhibit different or slightly similar outcomes, depending on the strain used, the dosage, and genetic factors. Here are some examples:

In the field of depression, psychobiotics emerge as a promising option that could complement or serve as an alternative to conventional treatments [12]. This potential is due to their influence on the intestinal microbiota and the metabolism of neurotransmitters such as tryptophan, which is crucial in the production of serotonin. In a study conducted by Tian et al., 2022 [34], the effect of *Bifidobacterium breve* CCFM1025 on patients with Major Depressive Disorder (MDD) was explored, showing significant improvements in depression and positively impacting the intestinal microbiota and tryptophan metabolism. This finding suggests that modifying the intestinal microbiota through psychobiotics could alter serotonin production, providing a novel mechanism for antidepressant treatment.

Additionally, Kazemi et al., 2019 [37] also found significant benefits using a combination of *Lactobacillus helveticus* and *Bifidobacterium longum*. This study demonstrated a notable reduction in the Beck Depression Inventory score compared to a placebo, supporting the hypothesis of the gut–brain axis as a relevant pathway in depression. The tryptophan modulation favored by these probiotics suggests an enhanced production of serotonin, directly impacting depressive symptoms.

Komorniak et al., 2020 [54] explored the impact of *Lactobacillus plantarum* 299v on patients with depression. Their findings indicate improvements in both depressive symptoms and inflammatory markers, suggesting that this probiotic could strengthen the integrity of the intestinal barrier and reduce inflammation. This, in turn, modulates the transmission of neurotransmitters through the gut–brain axis, potentially improving brain function and mood.

Meanwhile, Miyaoka et al., 2018 [50] assessed the adjunctive use of Clostridium butyricum MIYAIRI 588 in cases of treatment-resistant depression. They observed significant improvements in the depression indices of the patients when this probiotic was combined with standard antidepressant therapy. The researchers suggest that the improvement is due to the production of short-chain fatty acids, such as butyrate, which have neuroprotective and anti-inflammatory properties.

Ullah et al., 2022 [44] investigated the effects of probiotic supplementation on the levels of Brain-Derived Neurotrophic Factor (BDNF) and its relationship with the mitigation of depressive symptoms. Their results showed an increase in BDNF levels, accompanied by a general improvement in symptoms, supporting the hypothesis that probiotics have significant therapeutic potential in the treatment of depression.

Schneider et al., 2023 [43] focused on studying the impact of high-dose probiotic supplementation on cognition and brain functions in patients with MDD. They reported notable improvements in verbal episodic memory and in brain activation, particularly in the hippocampus. This implies that psychobiotics could enhance cognitive function through the modulation of inflammatory responses and the elevation of neurotrophic factors like BDNF.

Finally, Baião et al., 2023 [55] discovered that multispecies probiotics reduce the prominence of negative emotional stimuli and significantly improve mood scores. This provides a contrast to traditional antidepressants, which generally optimize emotional biases towards positive stimuli. The study suggests that this action could be through the modulation of central GABA neurotransmission, a mechanism not directly influenced by traditional antidepressants. Research on psychobiotics for depression reveals significant variability in the strains used and their mechanisms of action.

In the context of cognitive impairment and more complex psychiatric disorders such as schizophrenia and bipolar disorder, the role of psychobiotics can be particularly nuanced. Cognitive impairments, often linked to neurodegeneration, might benefit from the neuroprotective or neuroplastic potential of psychobiotics [24,60]. However, the multifactorial nature of schizophrenia and bipolar disorder, involving genetic, environmental, and neurobiological factors, poses a significant challenge [67,69]. These conditions suggest that the gut–brain axis is a promising yet complex target for intervention, where psychobiotics could modulate immune responses and neurotransmitter levels beneficially, though their impact might vary greatly among individuals [30].

Furthermore, while depression shows considerable promise in responding to psychobiotic interventions due to more definitive links between gut health and mood regulation, the complexities increase with disorders like schizophrenia and bipolar disorder [21,87]. These conditions require a more intricate exploration of how psychobiotics influence them, suggesting that some mental health conditions may be more receptive to psychobiotic treatment than others, which might necessitate more sophisticated interventions.

This differential impact across various disorders not only highlights the potential of psychobiotics to contribute uniquely to the management of distinct psychiatric conditions but also emphasizes the critical need for personalized treatment approaches [23,25]. Careful, evidence-based selection of probiotic strains and their application in appropriate doses are fundamental to maximizing the therapeutic benefits of psychobiotics, adapting interventions to the needs and biological specifics of each patient [24,88].

The variability of disorders evaluated also allowed for the identification of common findings related to the frequency of strains used, and these studies highlight the significant role of strains such as *Lactobacillus helveticus* R0052, *Lactobacillus plantarum*, and *Bifidobacterium longum* in modulating various health conditions, both mental and physical. For example, *Lactobacillus helveticus* R0052 has been widely used in various contexts [38,74,76]. In the area of stress and anxiety, this strain is used in combination with Bifidobacterium longum in products like OMNi-BiOTiC^®^ Stress Repair to help repair intestinal function affected by stress, which can indirectly improve psychological symptoms [76]. Regarding digestive health, it contributes to the improvement of digestion and nutrient absorption, as well as balancing the intestinal flora. Additionally, studies suggest it can enhance cognitive health due to its impact on the gut–brain axis [89].

On the other hand, *Lactobacillus plantarum* is available in formulations of up to 1.5 × 10¹⁰ CFU twice a day and is used in various disorders. For digestive health, it is commonly used to treat and prevent diarrhea, including that associated with antibiotic use and traveler’s diarrhea. In treating irritable bowel syndrome, certain strains are specifically studied to improve intestinal motility and reduce inflammation. It can also help lower LDL cholesterol and improve blood pressure, benefiting cardiovascular health. Moreover, it has the potential to improve allergic responses and reduce the severity of autoimmune diseases such as eczema [76].

Finally, *Bifidobacterium longum* is present in formulations of up to 10 × 10⁹ CFU/day and plays a crucial role in several health aspects. In combination with *Lactobacillus helveticus*, it is used to improve mental well-being and reduce the stress response in contexts of stress and anxiety [76]. It promotes a healthy digestive system and aids in the treatment of diarrhea and irritable bowel syndrome, strengthening digestive health. Furthermore, it boosts the immune system, potentially reducing the incidence and duration of common colds, thereby enhancing immune function [25,76].

### 4.3. Efficacy of Psychobiotics in Treating Cognitive and Psychiatric Disorders

The use of probiotics, predominantly from the Lactobacillus and Bifidobacteria families, over treatment durations ranging from 4 to 24 weeks [45,47,52], underscores the need for an in-depth understanding of how different strains and combinations could optimize therapeutic outcomes.

The dosage of probiotics plays a critical role in their efficacy. An insufficient dose may not result in significant changes in the intestinal microbiota that reflect clinical improvements. Determining the optimal dose should be based on studies that evaluate the relationship between the amount of probiotics administered and changes in clinical and biochemical markers of depression, including understanding the minimum effective dose and the possible therapeutic ceiling, where additional doses do not result in incremental benefits [90,91]. Studies such as those by Tian et al., 2022 [34] and Kazemi et al., 2019 [37] have shown significant antidepressant effects with strains like *Bifidobacterium breve* CCFM1025 and combinations of *Lactobacillus helveticus* with *Bifidobacterium longum*.

The unique pathophysiology of cognitive decline, schizophrenia, and bipolar disorder involves complex, multifactorial mechanisms that may benefit from tailored psychobiotic strategies. For instance, neuroprotective psychobiotics might be advantageous for cognitive decline, whereas those modulating neuroimmune and neuroendocrine pathways could be more suitable for schizophrenia and bipolar disorders [87,92,93].

The response to psychobiotic treatment can vary significantly, suggesting that a personalized medicine approach is crucial. This individualization is necessary due to the distinct interactions between psychobiotic strains and the host’s gut microbiota, which can influence the efficacy of the treatment [94,95].

To fully establish the effectiveness of psychobiotics across a broad spectrum of psychiatric conditions, extensive and rigorous clinical trials are needed. These studies should aim to determine the optimal doses, identify the most effective strains, and clarify the mechanisms through which psychobiotics exert their effects [25].

Reininghaus et al., 2020 [36] and Romijn et al., 2017 [38] both illustrate that while psychobiotics significantly improve psychiatric symptoms over time, comparative studies between different groups using varying scales like HDRS and BDI-II show no significant differences, pointing to the complexity of measuring psychobiotic efficacy.

Sacarello et al., 2020 [39] demonstrated that *Lactobacillus plantarum* HEAL9 can produce rapid and clinically relevant improvements in depressive symptoms within just two weeks. That study highlights the robust efficacy of psychobiotics, particularly Lactobacillus, in swiftly influencing mood disorders, thereby underscoring their potential as a powerful component in the management of depression.

Studies like those by Ullah et al., 2022 [44] and Schneider et al., 2023 [43] show that apart from improving depressive symptoms, psychobiotics can enhance cognitive functions, possibly through the modulation of inflammatory responses and neurotrophic factors like BDNF.

Regarding cognitive decline, various studies have demonstrated the usefulness of strains in improving specific cognitive functions:*Bifidobacterium breve A1*: Xiao et al., 2020 [58] reported significant improvements in the total RBANS score, particularly in the immediate memory and visuoconstructive domains, after 16 weeks of supplementation.*Lactobacillus plantarum OLL2712*: Sakurai et al., 2022 [59] observed improvements in composite memory and visual memory in older adults, which may indicate a neuroprotective and anti-inflammatory effect of this strain.

When comparing results across different disorders and studies, it is evident that the efficacy of psychobiotic strains can be highly specific to the disorder and symptoms. For example, while *Bifidobacterium breve CCFM1025* is effective in reducing depressive symptoms and gastrointestinal problems, *Lactobacillus plantarum* appears to be particularly useful in enhancing cognitive functions in cognitive decline [34,76,77,83]. This suggests that the choice of strain should be carefully considered depending on the specific disorder and therapeutic goals.

These findings advocate for a personalized approach in psychobiotic administration and emphasize the necessity for targeted research to optimize these interventions, considering the heterogeneity in patient responses and specific biochemical pathways affected by different probiotic strains. Future research should focus on standardizing psychobiotic formulations and exploring their synergistic effects with conventional antidepressants to fully harness their therapeutic potential [14,24,96].

### 4.4. Heterogeneity in Study Designs and Evaluation Methods

Research in the field of psychobiotics for treating cognitive and psychiatric disorders is notable for its diversity, not only in terms of the probiotic strains investigated and the clinical conditions addressed but also in the heterogeneity of study designs and evaluation methods used. This variability poses particular challenges for synthesizing and comparing findings across studies, especially when considering consistency in the assessment scales used to measure depression and cognitive function [97,98].

Depression and cognition scales, such as the HDRS, the BDI, the PANSS for schizophrenia, and the RBANS, are crucial tools for assessing the efficacy of treatments. However, the choice of these scales varies considerably among studies, which can influence the interpretation of results and the ability to make direct comparisons between research [98].

For example, Tian et al., 2022 [34] reported improvements using BPRS, MADRS, and HDRS-24, covering a broad spectrum of depressive symptoms from somatic to cognitive and showing comprehensive benefits. In contrast, Chahwan et al., 2019 [35] used BDI-II, focused more on the cognitive aspects of depression, which might not fully capture the change in somatic symptoms.

Similarly, in studies targeting cognitive impairments, Xiao et al., 2020 [58] utilized RBANS, which assesses broad cognitive domains and showed improvements in memory and visuospatial skills. On the other hand, Sakurai et al., 2022 [59] used more targeted scales like MPI, VIM, and VBM to demonstrate improvements in memory and visual–spatial integration, offering a detailed view of the specific cognitive benefits of psychobiotics.

This scale variability can significantly affect outcomes and interpretations. For instance:Reininghaus et al., 2020 [36] used HDRS and BDI-II and noted improvements, but without a control group performing differently, ’it is challenging to attribute changes directly to the psychobiotics due to potential placebo effects.Romijn et al., 2017 [38], employing MADRS, DASS-42, and QIDS-SR16, found no effective treatment for low mood, which could be attributed to the scales’ focus on different symptoms that might not align perfectly with psychobiotic effects.

The use of different scales across studies, like HDRS for more somatic-focused assessments versus BDI for cognitive–affective symptoms, can lead to results that seem contradictory or inconclusive when, in fact, they may be capturing different facets of psychobiotic efficacy [98].

This heterogeneity in assessment methods underscores the need for a more standardized approach in psychobiotic research, to allow for more direct comparisons and meta-analyses. The selection of scales should consider not only their validity and reliability but also how their specific features align with the study’s goals and hypotheses about the mechanisms of action of psychobiotics. Furthermore, clarity in presenting results and interpreting what constitutes a clinically significant change is essential to advance our understanding of the efficacy of psychobiotics in treating cognitive and psychiatric disorders. Adopting common guidelines for evaluation and result presentation could facilitate this process, allowing for a more coherent synthesis of the available evidence and enhancing our ability to determine the true efficacy of psychobiotic interventions. This approach clarifies that systematic reviews are essential for such purposes [99].

### 4.5. Safety, Tolerance, and Mechanisms of Action of Psychobiotics

One aspect to consider with psychobiotics is their safety and tolerance, as well as the mechanisms through which their effects are mediated. The studies evaluated in this review have demonstrated generally favorable safety profiles, but also highlight the importance of monitoring for adverse effects.

From a safety perspective, this review reveals a general pattern of good tolerance towards psychobiotics, with a spectrum of side effects ranging from mild to moderate and rarely serious. The scarcity of serious adverse events and the generally good tolerance highlight psychobiotics’ potential as a safe therapeutic option. However, despite these positive findings, this review also calls attention to areas needing improvement in future research, particularly concerning the risk of bias. The variability in terms of incomplete outcome data and selective reporting necessitates a cautious interpretation of the findings.

Adverse events reporting in psychobiotic trials varies, with gastrointestinal effects being the most common but typically mild, such as constipation and nausea reported by Romijn et al., 2017 [38], and mild gastrointestinal complaints like bloating and flatulence noted by Ghorbani Z et al., 2018 [49], and Vaghef E et al., 2023 [47]. These studies and others, including neurological and allergic reaction reports, suggest that while psychobiotics are generally well-tolerated, they do require careful monitoring for adverse effects, particularly in sensitive individuals.

The safety profiles emphasized by studies such as those by Dickerson F et al., 2014 [67] and Xiao et al., 2020 [58], which reported no serious adverse events, corroborate the safety of psychobiotics in clinical use. Yet, instances like the serious event of erectile dysfunction reported by Hwang Y et al., 2019 [61] underscore the necessity for diligent adverse event monitoring in clinical trials.

Regarding the mechanisms of action, psychobiotics exert beneficial effects through several pathways:Gut–Brain Axis Modulation: They primarily influence the gut–brain axis, involving neurotransmitter systems, immune responses, and inflammatory pathways that affect brain function and behavior [25].Anti-inflammatory Effects: Many psychobiotics reduce inflammation, which is implicated in psychiatric disorders such as depression and anxiety, as noted in studies like that by Hwang Y et al., 2019 [61].Neurotransmitter Modulation: Certain strains are involved in neurotransmitter production or modulation, which significantly impacts mood and cognitive functions [100].

This study aligns with prior research, like that by Dinan et al., 2013 [101], highlighting the role of psychobiotics in producing neuroactive substances such as gamma-aminobutyric acid and serotonin, thereby acting on the gut–brain axis to improve cognitive disorders. Furthermore, the review by Roy et al., 2023 [102] elucidates the role of psychobiotics in modulating central nervous system functions through immunological, neuronal, and metabolic pathways, demonstrating their strong antidepressant and anxiolytic potential.

### 4.6. Evaluation of the Evidence

The reviewed studies exhibit several limitations that affect the reliability and validity of their findings. Many, such as those by Tian P et al., 2023 [52] and Majeed M et al., 2018 [48], involve relatively small sample sizes, with intervention groups ranging from 15 to 20 participants. Such small sizes may limit the generalizability of the findings and also diminish the statistical power of the studies, making it challenging to detect meaningful effects that could be applicable to a wider population [103,104].

Additionally, the intervention periods in several studies were notably short. For instance, studies by Schneider et al., 2023 [43] and Dickerson F et al., 2014 [67] lasted only four weeks. This duration could be insufficient to fully observe the therapeutic potential of psychobiotics or to assess their long-term safety and effects, which might result in an incomplete understanding of their efficacy.

There is also significant variability in the types of probiotics used across different studies, as well as their formulations (including freeze-dried powder, capsules, and drinks) and doses. Romijn et al., 2017 [38] and Kazemi et al., 2019 [37] used different strains and formulations, making it challenging to directly compare their efficacy and identify which components are most effective, and this also complicates direct comparisons across studies and can impact the consistency of results. Such discrepancies may lead to conflicting interpretations of how effective psychobiotics truly are.

In-depth analysis of the mechanisms through which psychobiotics act is also crucial and should be a focus of future studies. This includes detailed investigations into how these interventions affect neurotransmitter systems, immune responses, and inflammation pathways, which are believed to play critical roles in the gut–brain axis.

Another critical issue is the inconsistency in reporting and potential methodological biases. Studies like that of Reininghaus et al., 2020 [36] highlight significant improvements in psychiatric symptoms over time; however, the lack of significant differences between the treatment and control groups calls into question the efficacy of the intervention. This could be attributed to selective reporting or incomplete data presentation, which can introduce bias and affect the study’s credibility.

Given these issues, future research should extend the duration of trials to provide more comprehensive data on the efficacy and safety of psychobiotics, allowing for observations of longer-term effects and the full therapeutic potential of these treatments. Utilizing standardized and validated scales across studies would facilitate more accurate comparisons and enhance meta-analytical processes. Increasing the sample sizes and including a more diverse range of participants can improve the reliability and generalizability of the findings. Larger groups would provide greater statistical power, while a diverse sample structure would ensure the findings are applicable across different demographics and physiological profiles [105].

Researchers should strive for transparency and comprehensiveness in the reporting of their methodologies. Detailed accounts of randomization processes, blinding methods, and full outcome data reporting should be standard practices. This level of detail would help minimize potential biases and improve the overall quality of the research. Implementing these recommendations would significantly strengthen the field of psychobiotic research, providing clearer insights into their potential as therapeutic agents for mental health and cognitive disorders. The aim would be not only to ascertain the efficacy of psychobiotics but also to understand their mechanisms and optimize their use in clinical settings.

### 4.7. Clinical and Preclinical Evidence of Benefits

The impact of psychobiotics on mental health and neuropsychiatric disorders offers a deeper understanding of how changes in the intestinal microbiota can influence these conditions. This understanding highlights the significance of the intestinal microbiota in regulating emotional and cognitive processes, and how its modulation through psychobiotics could represent an alternative and promising therapeutic strategy. However, although initial studies are promising, the execution of large-scale controlled studies is crucial to confirm these beneficial effects and to better understand the underlying mechanisms through which psychobiotics can influence mental and cognitive health. Research in this field is in its early stages but provides an intriguing path toward the development of new therapies for psychiatric and cognitive disorders based on the modulation of the intestinal microbiota [20,106,107].

Misra and Mohanty [12] offer compelling evidence that supports the role of psychobiotics in managing psychiatric and cognitive disorders. Both pieces of research highlight the potential of psychobiotics to act as a novel therapeutic pathway, particularly for conditions such as depression, anxiety, cognitive impairments, and stress-related disorders. Misra and Mohanty focus on the influence of the gut microbiome on mood and cognition. They propose that an increase in beneficial gut bacteria can diminish inflammation, lower cortisol levels, and alleviate symptoms of depression and anxiety. Their review further explores the mechanisms through which psychobiotics might exert their effects, including the production of neurochemicals, modulation of the gut–brain axis, anti-inflammatory actions, and effects on the hypothalamic–pituitary–adrenal (HPA) axis.

Misra [12] and this study agree on the significance of the gut–brain axis as a critical mediator of the therapeutic effects of psychobiotics. This study offers a more detailed analysis of clinical outcomes in human trials, presenting evidence of psychobiotics’ efficacy across a broader spectrum of psychiatric and cognitive disorders. In contrast, Misra and Mohanty’s review delves into the potential underlying mechanisms, including neurochemical and immunological pathways, which may mediate the observed benefits.

Another study supporting the positive outcomes of psychobiotics is the research by Coelho and Kerpel in 2022 [108]. This study focuses on depression and provides a systematic review that supports the use of psychobiotics as a supplement to conventional treatment with antidepressants for major depressive disorder. Its findings reveal significant improvements in depression assessments through psychiatric scales, indicating a reduction in anhedonia, cognitive reactivity, and insomnia in patients affected by this disorder. Furthermore, significant changes were observed in aspects related to the pathogenesis of depression, such as dysbiosis and inflammation.

Similarly, Sarkar et al., 2016 [22] reported positive trends in treating depression, anxiety, cognitive impairment, and other conditions. They also highlight the need to explore the response to different doses, long-term effects, and the expansion of the concept of psychobiotics beyond traditional probiotics and prebiotics, to include any exogenous influence that can affect the brain through bacterial mediation, variables mentioned in this work.

When comparing the results of this study with those of Correll et al., 2015 [109] regarding the risks of physical illnesses in people with schizophrenia, depression, and bipolar disorder due to the use of psychotropic drugs, this study emphasizes that, although psychotropic medications are essential for treating severe mental disorders, they are also associated with an increased risk of various physical diseases, including obesity, dyslipidemia, diabetes mellitus, thyroid diseases, hyper- or hyponatremia, cardiovascular diseases, respiratory tract diseases, gastrointestinal diseases, hematological diseases, musculoskeletal diseases, and renal diseases, as well as movement disorders and seizures. In contrast, the adverse effects of psychobiotics found in this study could be considered potentially less severe or different from those caused by conventional antipsychotics, antidepressants, and mood stabilizers. However, it is crucial to note that the existing literature on the adverse effects of psychobiotics is limited, and more research is required to fully understand their safety profile compared to conventional treatments for mental disorders.

Another important point to consider is that the most frequently found species identified as psychobiotics in this study, specifically, *Lactobacillus helveticus* and *Lactobacillus plantarum*, are commonly mentioned. These strains are used in various studies, indicating their popularity and potential importance in modulating psychobiotic effects. *Bifidobacterium longum* and *Bifidobacterium breve* were also frequently found. These microorganisms are known for their benefits to intestinal health, which may be related to their positive effects on mental health.

Liu et al., 2018 [110] explores the multifaceted roles of *Lactobacillus plantarum* as a probiotic, particularly its effects on the gut–heart–brain axis. It emphasizes the probiotic’s potential in managing inflammatory bowel diseases, metabolic syndromes, dyslipidemia, hypercholesterolemia, obesity, diabetes, and psychological disorders. The mechanisms include modulation of gut microbiota, reduction of inflammation, and improvement in metabolic functions. *L. plantarum*’s ability to navigate through the gastrointestinal tract and adhere to intestinal epithelial cells, and its competitive inhibition of pathogens, highlight its probiotic characteristics and support its use in enhancing gut health and beyond.

Cebeci and Gurakan’s 2003 [111] study evaluated fifteen Lactobacillus strains for probiotic properties, primarily focusing on Lactobacillus plantarum. Key findings included the following:Tolerance to acid and bile salts, crucial for surviving the gastrointestinal environment.Ability to ferment fructooligosaccharides (FOS), beneficial for gut health.β-galactosidase activity, important for lactose digestion.Antibiotic susceptibility, relevant for safety and therapeutic use.

Arasu et al., 2016 [112] explores the in vitro significance of the probiotic *Lactobacillus plantarum* in the medical field, highlighting its broad applications due to its antioxidant, anticarcinogenic, anti-inflammatory, antiproliferative, anti-obesity, and anti-diabetic properties. The medical applications of *L. plantarum* were investigated, emphasizing its potential to treat chronic and cardiovascular diseases such as Alzheimer’s, Parkinson’s, diabetes, obesity, cancer, and hypertension without adverse side effects.

Seddik et al., 2017 [113] highlights the probiotic properties and food applications of Lactobacillus plantarum. It underscores its adaptability and ability to produce bacteriocins, antimicrobial compounds that offer applications in both food preservation and as supplements to antibiotic treatments. That review emphasizes the safety of *L. plantarum* endorsed by health authorities and presents studies supporting its beneficial use in preventing gastrointestinal disorders, managing cholesterol, and treating irritable bowel syndrome. The importance of genomic analyses to better understand the probiotic functionality of *L. plantarum* and its genetic diversity is stressed, underlining its potential as a versatile biological agent with promising applications in medical, veterinary, and food fields.

Behera et al., 2018 [114] focuses on *Lactobacillus plantarum,* highlighting its significant role in enhancing the safety and extending the shelf-life of fermented foods. It emphasizes *L. plantarum*’s widespread use as a probiotic and microbial starter in the food industry due to its beneficial effects against harmful microflora and its ability to improve nutritional and technological features of foods and beverages. Behera et al. also discuss the strain’s identification in traditional foods, its enzyme systems, and the production of bioactive compounds like bacteriocin, showcasing the potential of *L. plantarum* strains to contribute positively to food safety and quality.

The study of Zago et al., 2011 [115] evaluated the probiotic potential of *Lactobacillus plantarum* strains isolated from cheeses, focusing on resistance to biological barriers like lysozyme, bile, and simulated gastric juice, as well as bile salt hydrolase (BSH) activity and surface hydrophobicity. It found variability in resistance levels and hydrophobicity among strains, with some demonstrating significant potential for use in probiotic foods. The research emphasizes the importance of selecting strains with robust traits for developing effective probiotic products.

The study by Xiao et al., 2020 on *Bifidobacterium breve* and its impact on the cognitive functions of older adults with suspected mild cognitive impairment showed promising results. By administering the probiotic *B. breve* A1 for 16 weeks to physically healthy subjects with suspected mild cognitive impairment, a significant improvement was observed in the total RBANS score, which assesses cognitive functions, particularly in immediate memory, visuospatial/constructive abilities, and delayed memory, with no adverse events reported. This suggests that *B. breve* A1 is a safe and effective approach to enhancing memory functions in this population [58].

Toscano et al. in 2015 [116] focused on evaluating the probiotic characteristics of *Bifidobacterium breve* M-16V, *B. longum* subsp. *infantis M-63*, and *B. longum* subsp. *longum BB536*. These strains were assessed for their growth compatibility, resistance to antibiotics, antibacterial activity against pathogens, resilience against gastric acidity, bile salt hydrolysis, and adhesion to human intestinal epithelial cells. The findings highlight that *B. breve M-16V* showed significant antibacterial activity, and all strains demonstrated strong adhesion to HT29 cells and showed resistance to gastric acidity, making them promising candidates for probiotic use.

Okubo et al., 2019 [117] explores also the impact of *Bifidobacterium breve A-1* on anxiety and depressive symptoms in individuals with schizophrenia, examining its effects on immune products like cytokines and chemokines. The research findings indicate a potential therapeutic role for this probiotic in improving psychiatric symptoms, suggesting that further investigation into its benefits across different psychiatric conditions and its interaction with dietary habits and gut microbiome composition is warranted.

Cionci et al., 2018 [118] discusses the therapeutic role of *Bifidobacterium breve* as a dietary supplement for children’s health, highlighting its beneficial effects in various pediatric conditions. It covers the organism’s antimicrobial capabilities, lack of transmissible antibiotic resistance, non-toxic nature, and immunostimulatory abilities. *B. breve* has shown promise in treating diarrhea, infant colic, celiac disease, obesity, allergies, and even as a support during chemotherapy or antibiotic treatments, emphasizing its role in the emerging field of therapeutic microbiology.

Considering the strains of psychobiotics discussed, it is clear that Lactobacillus helveticus, *Lactobacillus plantarum*, *Bifidobacterium longum*, and *Bifidobacterium breve* stand out due to their frequent mention and broad application in various studies. These strains not only show potential in improving mental health but also exhibit a wide range of benefits for physical health, demonstrating their versatility and importance.

### 4.8. Future Applications and Administration Technology

The administration of psychobiotics faces key challenges related to viability, efficacy, and stability, primarily due to these microorganisms’ sensitivity to adverse gastrointestinal conditions, stability and storage issues, and the untargeted release of active compounds. These factors can compromise the efficacy of psychobiotics before they reach the intestine, where they exert their beneficial effect. Conventional forms of administration, such as tablets and capsules, although popular for allowing precise dosing and being easy to consume, face these drawbacks, limiting their practicality and accessibility [119,120,121,122].

In this context, the doses used in the studies of this systematic review varied widely, from 1 × 10^9^ to 1 × 10^10^ CFU per day, suggesting that a range of 1 × 10^9^ to 3 × 10^9^ CFU per day could be a common starting point for future research. However, optimizing dosage for each type of disorder and specific probiotic strain still requires further research to effectively personalize psychobiotic treatments [123].

In terms of forms of administration, lyophilized powder, capsules, and tablets are the most common, chosen for their ability to preserve beneficial bacteria, allow precise dosing, and facilitate consumption. Lyophilization is particularly useful for maintaining the stability of probiotics, while capsules and tablets offer convenience and efficacy. Although less common, fermented drinks present an attractive alternative for those who prefer a liquid form or have difficulty swallowing capsules [88].

The choice of formulation is based on factors such as probiotic stability, ease of administration, and individual preferences, highlighting the importance of considering patients’ needs and preferences when designing treatments. However, given the limitations of traditional administration forms, nanotechnology emerges as a promising solution, offering new ways for the encapsulation and delivery of probiotics with greater efficiency, precision, and the overcoming of gastrointestinal challenges, potentially improving the viability and efficacy of psychobiotics [124,125,126,127].

Nanotechnology, through the creation of nanomaterials and nanoencapsulation techniques, presents innovative solutions to overcome the limitations of traditional probiotic administration methods. These advanced technologies include the following [124,128]:Enhanced Viability and Efficacy: Nanoencapsulation protects sensitive microorganisms from adverse gastrointestinal conditions, improving survival and enabling effective delivery to the intestine.Targeted Delivery: Advanced nanotechnology techniques allow for the targeted release of psychobiotics to specific tissues, including the brain, overcoming biological barriers such as the blood–brain barrier and opening new avenues for treating neurological diseases.Innovations in Nanostructured Materials: The use of diverse nanostructured materials, such as nanocellulose, magnesium oxide nanoparticles, and chitosan nanoparticles, offers unique advantages in terms of biocompatibility, mechanical stability, and thermal resistance, enhancing the efficacy of psychobiotics.Applications in Oral and Gastrointestinal Health: Nanoencapsulation of psychobiotics shows promising potential in treating oral and gastrointestinal diseases, offering effective protection against pathogens and improving colon health.

Despite significant advances in nanotechnology applied to psychobiotic delivery, challenges remain that must be addressed to ensure the safety and efficacy of these innovations, including the following [123,125,129]:Nanomaterial Toxicity: Comprehensive evaluation of the biocompatibility and potential toxicity of nanomaterials is essential to ensure they are safe for human use.Need for Further Research: More research and clinical trials are crucial to optimize nanoencapsulation formulations, evaluate their safety, and confirm their efficacy in various medical applications.

Relevant to the future of psychobiotics is the integration of bioinformatics, synthetic biology, and artificial intelligence for species enhancement, the development and bioprospecting of new probiotics, and the identification of peptide sequences that perform immunomodulatory, antimicrobial, and gut microbiota regulation functions, among others.

At the forefront of contemporary science, the integration of artificial intelligence (AI), synthetic biology (SB), and bioinformatics is revolutionizing intestinal microbiome research. This powerful combination of technologies offers new dimensions for understanding and manipulating this complex ecosystem, with significant implications for human health. Using machine learning algorithms and bioinformatic techniques, in-depth analysis of complex biological interactions can be performed. Simultaneously, SB opens innovative pathways for designing probiotics with enhanced functionalities, promising to transform both our understanding and therapeutic approach to chronic, inflammatory, and neurological diseases [130,131].

Traditionally, approaches to developing and administering probiotics have been limited by the capacity to analyze and manipulate the intestinal microbiome precisely. These limitations result in ineffective personalization of treatments, a limited understanding of the microbiome’s diversity and complexity, and slow development of new probiotic strains with specific therapeutic properties. However, the adoption of AI and bioinformatics in this field is overcoming these barriers, enabling accurate disease prediction and the personalization of therapies based on individual microbiotic profiles. This is achieved through the analysis of large data sets and the identification of complex patterns, revealing how probiotic supplementation could favorably alter the microbiome composition [132,133].

Furthermore, SB, supported by bioinformatic analysis, is facilitating the creation of probiotics and bacteriophages with new functionalities. Techniques like CRISPR-Cas genome editing, greatly benefited by bioinformatic tools, are revolutionizing the ability to precisely modify microbial genomes. Additionally, innovative studies like ABIOME demonstrate how the combination of AI, SB, and bioinformatics can optimize probiotic therapies. Through machine learning models, specific formulations that produce bioactive metabolites with potential therapeutic benefits are designed, highlighting the capability of these technologies to create more effective and personalized treatments [134,135].

The future of analyzing and manipulating the intestinal microbiome through AI, SB, and bioinformatics holds extraordinary promise, with the potential to revolutionize our understanding and management of a wide range of health conditions [136]. The continued development of more robust and generalizable methods, along with meticulous evaluation and regulatory authority approval for clinical use of these technologies, is essential. The convergence of these disciplines marks the beginning of a new era in personalized medicine, promising a future where optimizing intestinal and overall health can be done precisely and personally, marking a significant advancement toward more effective disease prevention and treatment [130,135].

## 5. Conclusions

This systematic review highlights the potential of psychobiotics, particularly strains of Lactobacillus and Bifidobacterium, as alternative therapies for managing psychiatric and cognitive disorders such as depression, cognitive impairment, schizophrenia, and bipolar I disorder. The findings from randomized controlled trials suggest that psychobiotics are generally safe and can offer significant benefits in symptom management. However, significant areas requiring improvement are identified, especially regarding the risk of bias from incomplete outcome data and selective reporting. It is emphasized that future studies need to adopt more rigorous methodologies, including improved randomization and blinding methods, to enhance the reliability of research findings.

Furthermore, it is suggested that future research should explore the diversity of treatment strategies to understand how different strains and combinations can optimize therapeutic outcomes for various conditions. There is a crucial need for larger and more diverse sample sizes to improve generalizability and statistical power.

## Figures and Tables

**Figure 1 nutrients-16-01352-f001:**
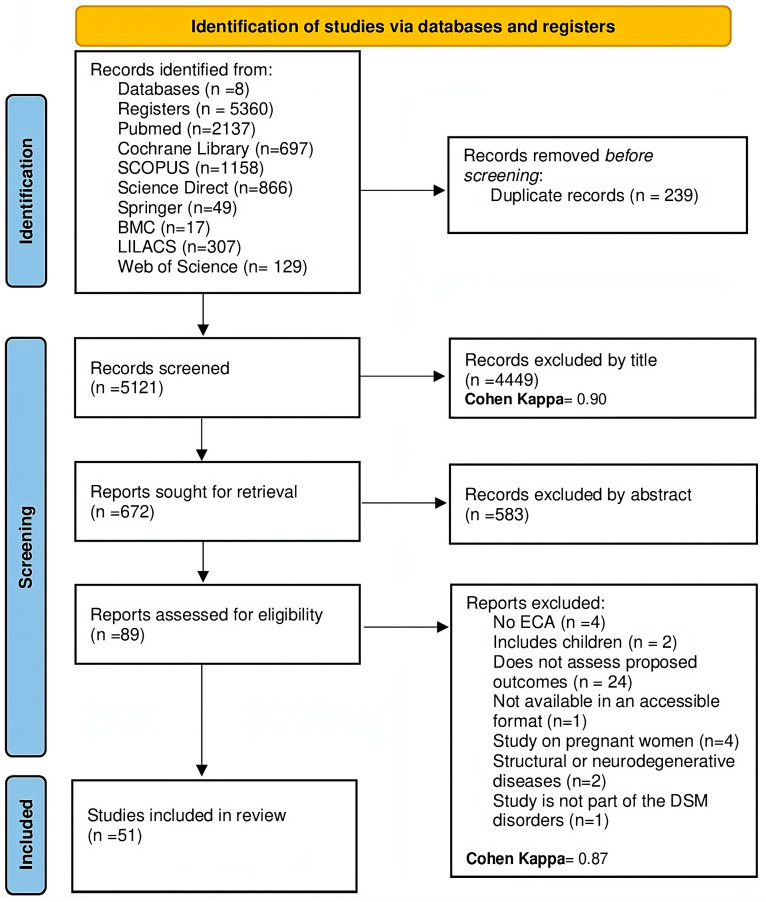
PRISMA flow diagram with the search and study selection strategy.

**Figure 2 nutrients-16-01352-f002:**
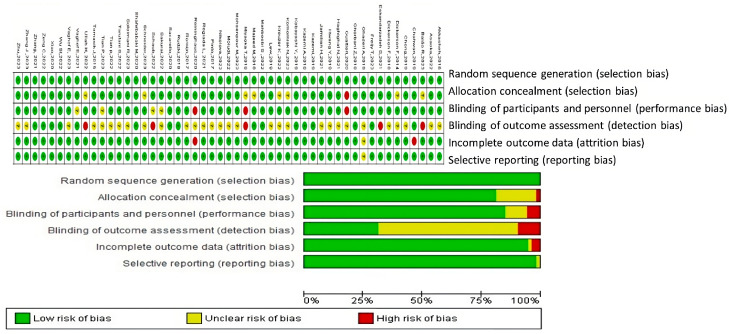
Risk of bias assessment for studies included in this systematic review. Green + indicates a low risk of bias. Studies with this symbol were considered to have a low likelihood of bias affecting their results. Red—represents a high risk of bias. Studies with this symbol were determined to have a high likelihood of bias that could significantly affect their results. Yellow ? denotes an unclear or unknown risk of bias. Studies with this symbol had insufficient information to determine the likelihood of bias [34,35,36,37,38,39,40,41,42,43,44,45,46,47,48,49,50,51,52,53,54,55,56,57,58,59,60,61,62,63,64,65,66,67,68,69,70,71,72,73,74,75,76,77,78,79,80,81,82,83,84].

**Table 1 nutrients-16-01352-t001:** Synthesis of the studies included in this systematic review.

Author, Year	Subjects (I/C)	Diagnosis	Average Age (Years)	Sex (%female)	Psychobiotic (Form; Dose)	Treatment Duration (Weeks)	Evaluated Outcome	Depression Scale Used	Conclusion
Tian et al., 2022 [34]	I: 20 C: 25	Depression	51.3	67	Freeze-dried powder; *B. breve* CCFM1025 10^10^ UFC	4	Improvement in depression score	BPRS, MADRS, HDRS-24	CCFM1025 is a promising psychobiotic that mitigates depression and associated gastrointestinal disorders.
Chahwan et al., 2019 [35]	I: 34 C:37	Depression	36.6	70	Freeze-dried powder; Ecologic ^®^Barrier 2.5 × 10^9^ UFC/g	8	Improvement in depression score	BDI-II	The consumption of probiotics can induce changes in cognitive patterns associated with depression.
Reininghaus et al., 2020 [36]	I:28 C:33	Depression	43	77	Freeze-dried powder; OMNi-BiOTiC^®^ Stress Repair 7.5 × 10^9^ organisms +125 mg of vitamin B7	4	Improvement of psychiatric symptoms	HDRS, BDI-II	Both groups significantly improved over time in psychiatric symptoms. However, there were no significant differences between groups in the scores of the scales used.
Kazemi et al., 2019 [37]	I1:38 I2:36 C:36	Depression	36.1	69	Freeze-dried powder; *L. helveticus R0052*, *B. longum* R0175 10 × 10^9^ UFC or Galactosaccharide	8	Improvement in depression score	BDI	Probiotic supplements in people with depression resulted in an improvement in the BDI score compared to placebo, while no significant effect was observed when prebiotics were used.
Romijn et al., 2017 [38]	I:40 C:39	Depression	35.8	78	Freeze-dried powder; *L. helveticus* R0052 and *B. longum* R0175 3 × 10^9^ UFC/day	8	Improvement in the scores of the scales used	MADRS, DASS-42, QIDS-SR16	Prescribing probiotics is not effective for treating low mood. The lack of observed effect on symptoms may be due to the severity, chronicity, or resistance to treatment of the sample.
Sacarello et al., 2020 [39]	I:45 C:44	Depression	48.6	73	Tablet; SAME 200 mg and *L. plantarum* HEAL9 1 × 10⁹ UFC/day	6	Improvement in depression score	Z-SDS	The combination of SAMe and L. plantarum HEAL9 in adults with mild to moderate depression symptoms produced clinically significant effects after 2 weeks.
Nikolova et al., 2023 [40]	I:24 C:25	Depression	32.5	80	Multispecies probiotic capsule (14 strain) with 2 × 10^9^ UFC. 4 capsules/day	8	Improvement in depression score	HDRS-17, HAMA	Depressive symptoms improved in both groups, with a more significant reduction in the probiotic group from week 4. On the other hand, overall, the treatment was well tolerated.
Zhang et al., 2021 [41]	I:38 C:31	Depression	45.8	63.7	Drink; *L. paracasei strain Shirota* 1 × 10^10^ CFU/mL/day	9	Improvement in depression score	HDRS, BDI	HDRS and BDI scores significantly decreased. The degree of depression improved in both the placebo and intervention groups; however, there were no significant differences between groups.
Akkasheh et al., 2016 [42]	I:17 C:18	Depression	38.3	85	Capsule; *L. acidophilus* 2 × 10^9^ CFU/g, *L. casei* 2 × 10^9^ CFU/g, and *B. bifidum* 2 × 10^9^ CFU/g/day	8	Improvement in depression score	BDI	After 8 weeks of intervention, patients who received probiotic supplements had a significantly lower total BDI score.
Schneider et al., 2023 [43]	I:30 C:30	Depression	38	60	Drink; Vivomix^®^ 900 × 10^9^ CFU/day	4	Improvement of cognitive symptoms	VLMT	Additional supplementation with probiotics improves verbal episodic memory and affects the neuronal mechanisms underlying cognitive impairment in depression.
Ullah H et al., 2022 [44]	I:36 C:32	Depression	39	58.4	*L. helveticus Rosell^®^-52*, *B. longum* Rosell^®^-175, 3 × 10^9^ CFU/day, vitamina B6 1.70 mg and SAME 200 mg/day	12	Improvement in depression score	HDRS, PHQ-9	Daily intake of SAMe and probiotic is effective in improving symptoms and quality of life in subjects with subthreshold depression and mild to moderate depression.
Tarutani S et al., 2022 [45]	I:9 C:11	Depression	53	85	Syrup; 4G-beta-D-galactosylsucrose 3.2 g	24	Improvement in depression score	MADRS	Intake of the prebiotic can improve self-efficacy, but not depressive symptoms, even in a small sample.
Vaghef E et al., 2021 [46]	I:22 C:23	Depression	37.4	NR	Freeze-dried powder; inulin 10 g/day	8	Improvement in depression score	HDRS, BDI-II	The use of prebiotic for 8 weeks does not significantly improve depression symptoms compared to the placebo group.
Vaghef E et al., 2023 [47]	I:22 C:23	Depression	38.5	NR	Freeze-dried powder; inulina 10 g/day	8	Improvement of symptoms	HDRS	In the short term, supplementation with prebiotics had no significant beneficial effects on depressive symptoms.
Majeed M et al., 2018 [48]	I:20 C:20	Depression	43.8	85	Tablets; *B. coagulans* MTCC 5856 2 × 10^9^ CFU/day	12	Improvement in depression score	HDRS, MADRS, CES-D	The probiotic showed solid efficacy in treating patients experiencing symptoms of irritable bowel syndrome with major depressive disorder.
Ghorbani Z et al., 2018 [49]	I:20 C:20	Depression	34.4	70	Capsule; Familact H^®^ 500 mg 2 capsules/day	6	Improvement in depression score	HDRS-17	The symbiotic group had a significantly lower HAM-D score compared to the placebo.
Miyaoka T et al., 2018 [50]	I:20 C:20	Depression	44.2	60	Tablets; *C.* *butyricum* MIYAIRI 588 60 mg/day	8	Improvement of symptoms	HDRS-17, BDI	The administration of probiotic in combination with antidepressants significantly improves symptoms of depression.
Kazemi A et al., 2018 [51]	I1:38 I2:36 C:36	Depression	36.7	71	Freeze-dried powder; *L. helveticus R0052* and *B. longum* R0175 10 × 10^9^ CFU/day; or galactooligosaccharide	8	Improvement in depression score	BDI-II	A beneficial effect of probiotics on the remission of depression was observed, as evidenced by an improvement in the BDI score. However, probiotics had no effect on the levels of circulating pro-inflammatory cytokines.
Tian P et al., 2023 [52]	I:15 C:13	Depression	38.8	NR	Freeze-dried powder; *B. breve* CCFM1025, *B. longum* CCFM687, and *P. acidilactici* CCFM6432 4 × 10^9^ CFU/g	4	Improvement in depression score	HDRS, MADRS, BPRS	Probiotic treatments can significantly mitigate psychiatric symptoms and comorbid gastrointestinal symptoms in patients with major depressive disorder.
Mohsenpour M et al., 2023 [53]	I:40 C:40	Depression	42.2	56	Drink; milk kefir tablets 500 cc/day	8	Improvement in depression score	BDI-II	The BDI-II score was reduced in both study arms; however, the comparison between groups was not statistically significant.
Komorniak N et al., 2023 [54]	I:21 C:17	Depression	44.9	NR	Capsule; Sanprobi Barrier^®^ 4 capsules 2 × 10^9^ CFU/day	5	Improvement in depression score	HDRS, BDI	An improvement in mental functioning of patients (reduction of BDI and HDRS) was evidenced, but it was not related to the probiotic used.
Baião R et al., 2023 [55]	I:35 C:36	Depression	27.9	63.4	Capsule; Bio-Kult^®^ 2 × 10^9^ CFU	4	Improvement of emotional processes	STAI, PHQ-9, ETB	The intake of probiotics decreased depression scores but did not correlate with changes in emotional processing.
Mahboobi S et al., 2022 [56]	I:39 C:35	Depression	38.9	78	Capsule; Probio-Tec^®^ *BG-VCap-6.5* at 1.8 × 10^10^ CFU *+* Magnesium chloride 500 mg. 2 capsules/day	9	Improvement in depression score	BDI-II	The administration of probiotic plus magnesium had no significant effects on mood, cognition, and intestinal integrity in individuals with obesity and depressed mood.
Kreuzer K et al., 2022 [57]	I:28 C:29	Depression	44.6	73.6	Freeze-dried powder; OMNi-BiOTiC^®^ Stress Repair 7.5 × 10^9^ organisms + vitamina B7	4	Improvement in depression score	HDRS, BDI-II	Both groups significantly improved in depression scores over time. However, no differences between groups were reported.
Xiao et al., 2020 [58]	I:40 C:39	Cognitive impairment	61.3	51	Capsule; *B. breve* A1, 1 × 10^10^ CFU 2 capsules/day	16	Improvement in RBANS score	RBANS	The total RBANS score significantly improved in the probiotic group after 16 weeks of consumption, particularly in the immediate memory and visuospatial/constructive domain.
Sakurai et al., 2022 [59]	I:39 C:39	Cognitive impairment	76.8	54	Freeze-dried powder; *L. plantarum* OLL2712 at 5 × 10^9^/day	12	Improvement in memory score	MPI, VIM, and VBM	Older adults who consumed the probiotic showed significant improvement in composite memory and visual memory compared to the placebo group.
Asaoka et al., 2022 [60]	I:55 C:60	Cognitive impairment	77.2	56	Freeze-dried powder; *B. breve* MCC1274 2 × 10^10^ CFU/day	24	Improvement in cognition	ADAS-Jcog and MMSE	According to the ADAS-Jcog subscale, orientation significantly improved compared to placebo at 24 weeks.
Hwang Y et al., 2019 [61]	I:50 C:50	Cognitive impairment	69.2	66	Capsule; fermented soy with *L. plantarum* C29 800 mg/day	12	Effects on cognition	VLMT, ACPT	Compared to the placebo group, the group that was administered probiotics showed greater improvements in combined cognitive functions, especially in the attention domain.
Kobayashi Y et al., 2019 [62]	I:61 C:60	Cognitive impairment	61.5	49.5	Capsule; *B. breve* A1 >2 × 10^10^ 2 capsules /day	12	Effects on cognition	RBANS, MMSE	At 12 weeks, neuropsychological test scores increased in subjects who consumed probiotic or placebo; however, no significant differences between groups were observed.
Ghaderi A et al., 2019 [63]	I:30 C:30	Schizophrenia	44.8	6	Tablet; probiotic 8 × 10^9^ CFU/day of *L. acidophilus*, *B. bifidum*, *L. reuteri* and *L. fermentum* (each one 2 × 10^9^) and 50,000 IU of vitamin D3 every 2 weeks	12	Improvement of symptoms	PANSS	The administration of probiotics plus vitamin D for schizophrenia for 12 weeks chronically had beneficial effects on the PANSS score.
Jamilian H et al., 2021 [64]	I:26 C:25	Schizophrenia	43.9	NR	Capsule; LactoCare^®^ 8 × 10^9^ CFU/day plus selenium 200 μg/day	12	Improvement of symptoms	PANSS	Co-supplementation with probiotics and selenium for 12 weeks in patients with chronic schizophrenia had beneficial effects on the overall PANSS score.
Tomasik J et al., 2015 [65]	I:30 C:27	Schizophrenia	44.8	35	Tablet; *L. rhamnosus* strain GG 10^9^ CFU and *B. animalis* subsp. lactis strain Bb12 10^9^ CFU/1 tablet/day	14	Effect on schizophrenia symptoms	PANSS	The provided probiotics had immunomodulatory effects, affecting molecules unresponsive to standard antipsychotic therapy. However, it did not reduce psychotic symptoms.
Soleimani et al., 2023 [66]	I:31 C:31	Schizophrenia	34.7	37	Capsule; FamiLact^®^ *Lactobacilo* 9 × 10^9^, *Bifidobacteria* 1.25 × 10^10^, and *S. Salivarius* 1.5 × 10^10^/day	12	Improvement of symptoms	BPRS, PANSS	Adding probiotics to oral antipsychotics did not improve psychiatric symptoms as measured through BPRS or PANSS.
Dickerson F et al., 2014 [67]	I:33 C:32	Schizophrenia	44.4	42	Tablet; *L. rhamnosus* strain GG 10^9^ CFU, *B. animalis* subsp. lactis Sib12 × 10^9^ CFU	14	Improvement of symptoms	PANSS	No significant differences were shown in the total PANSS score between probiotic supplementation and placebo.
Zeng C et al., 2022 [68]	I:21 C:21	TAB I	22.2	NR	Capsule; *Bifidobacterium*, *Lactobacillus*, and *Enterococcus* 1 × 10^7^ CFU, 6 capsules/day	12	Improvement of symptoms	YMRS, HAMA-14, HDRS	The symptom of mania was significantly alleviated in patients who received probiotic supplements compared to the placebo group.
ShahrbabakiM et al., 2020 [69]	I:19 C:19	TAB I	38.9	NR	Capsule; *B. bifidum*, *B. lactis*, *B. langum*, and *L. acidophilus* 1.8 × 10^9^ CFU, 1 capsule/day	8	Improvement of symptoms	YMRS, HDRS-17	Mania symptoms were significantly alleviated in patients who received probiotic supplements compared to the placebo group.
Dickerson F et al., 2018 [70]	I:26 C:26	TAB I	37.9	63	*L. rhamnosus* strain GG and *B. animalis* subsp. lactis strain Bb12 > 10^8^ CFU	24	Prevention of rehospitalizations	NR	The consumption of probiotics had no significant effects on the improvement and treatment of patients with bipolar disorder type 1.
Zhang J et al., 2023 [71]	I:46 C:44	TAB I-depressive	20.4	56	Freeze-dried powder; *B. animalis* subsp. *lactis* BAMA-B06/BAu-B0111, 1 × 10^9^ CFU/g–2 g/day	4	Improvement in depression score	HDRS-17, HAM-A-14	The use of probiotics is associated with a lower rate of rehospitalization in patients recently discharged after hospitalization for mania.
Gualtieri et al., 2020 [72]	I:65 C:32	Anxiety	43.8	61.9	Multispecies probiotic oral suspension ** 3 g/day.	12	Improvement in HAM-A score	HAM-A	Adjunctive therapy with probiotics may enhance the efficacy of conventional medications for bipolar I disorder, producing a favorable evolution of emotional state.
Eskandarzadeh S et al., 2021 [73]	I:24 C:24	Anxiety	34.1	81	Capsule; B. *longom*, *B. bifidum*, *B. lactis*, and *L. acidophilus* 18 × 10^9^ and sertraline 25 mg	8	Effects on severity of symptoms	HAM-A, BAI	The use of probiotic was associated with a decrease in the HAM-A scale score.
Pinto et al., 2017 [74]	I: 22 C: 22	Depression and anxiety	46.5	54.5	Freeze-dried powder; *B. longum* NCC3001 10^10^ CFU	6	Improvement in anxiety and depression score	HADS	The intake of the probiotic improved cognitive functions compared to placebo, in addition to aspects of mood and sleep.
Schaub et al., 2022 [75]	I:21 C:26	Depression and anxiety	39.4	57	Drink; Vivomix^®^ 90 × 10^9^ CFU/day	4	Improvement in anxiety and depression score	HDRS, BDI, STAI	The administration of the psychobiotic had beneficial effects on mild and moderate depression, improves quality of life, but does not reduce anxiety.
Zhu et al., 2023 [76]	I:30 C:60	Depression and anxiety	22	50	Freeze-dried powder; *L. plantarum* JYLP326 1.5 × 10^10^ CFU 2 times per day	3	Improvement in anxiety and depression score	HAMA-14, HDRS	Supplementary treatment with probiotics improves depressive symptoms and maintains a healthy enterotype.
Rudzki et al., 2018 [77]	I:30 C:30	Depression and anxiety	39	71.6	Capsule; *Lactobacillus Plantarum 299v* 10 × 10^9^ CFU 2 times per + ISSR	8	Improvement in depression, anxiety, and cognition score	HDRS, SCL-90, PSS-10	The administration of the probiotic JYLP-326 could significantly alleviate anxiety/depression symptoms and insomnia in university students anxious about exams.
Moludi et al., 2022 [78]	I:66 C:22	Depression and anxiety	51.2	39.5	Capsule; *L. Rhamnosus* 1.9 × 10^9^ and inulin 15 g/day	8	Improvement in anxiety and depression score	BDI, STAI	The administration of probiotic improved cognitive function in patients; however, there were no significant differences in the scores obtained from the HDRS and SCL-90 scales.
Haghighat N et al., 2021 [79]	I1:25 I2:25 C:25	Depression and anxiety	47	52	Capsule; synbiotic (15 g of prebiotics, 5 g of probiotic containing *L. acidophilus* T16, *B. bifidum* BIA-6, *B. lactis* BIA-7, and *B. longum* BIA-8 (2.7 × 10^7^ CFU/g each)) or probiotics (5 g of probiotics as in the synbiotic group) 4 times/day	12	Improvement in anxiety and depression score	HAD	Co-supplementation of probiotics and inulin for 8 weeks in patients with coronary artery disease produced benefits on depression, anxiety, and inflammatory biomarkers.
Regiada L et al., 2021 [80]	I1:27 I2:28 I3:26 C:25	Depression and anxiety	22.2	100	Capsules; probiotic composed of *Lactobacilllus* 20 × 10^9^ CFU (*L. acidophilus*, *L. plantarum*, *L. gasseri*, *L. paracasei*, *L. bulgaricus*, *L. brevis*, *L. casei*, *L. rhamnosus*, *L. salivaruys*) *Bifidobacteria* 10 billion CFU (*B. lactis*, *B. bifidum*, *B. breve*, *B. infantis*, *B. longum*); 1 capsule/day; or Probiotic + omega-3 200 mg/day; or omega-3 200 mg/day	12	Improvement in symptoms of depression, anxiety, and stress	BDI-II, STAI, PSS-10	12 weeks of supplementation with synbiotics produced an improvement in depression symptoms compared to probiotic supplementation in patients on hemodialysis.
Freijy T et al., 2023 [81]	I1:31 I2:28 I3:32 C:28	Depression, anxiety, and stress	36.3	91	Capsule; BioCeuticals^®^ 12 × 10^9^ CFU 2 times/day; or probiotic-rich diet 5 g/day; or combination of both (synbiotic)	8	Improvement in mood, depression, and anxiety score	BDI-II, pss-10, BAI	Omega-3, a probiotic, or a combination of both supplements did not reduce psychological symptoms in a sample of women in a non-clinical setting, compared to a placebo supplement.
Chong et al., 2019 [82]	I:56 C:55	Anxiety and stress	31.1	NR	Freeze-dried powder; *L. plantarum DR7* 1 × 10 ^9^ CFU/day	12	Improvement in anxiety, stress, memory, and cognitive function	PSS-10, DASS-42	A dietary intervention rich in prebiotics improves mood, anxiety, stress, and sleep in adults with moderate psychological distress and low prebiotic intake. A symbiotic combination does not appear to have a beneficial effect on mental health outcomes.
Lew et al., 2019 [83]	I:52 C_51	Anxiety and stress	31.3	77	Freeze-dried powder; *L. plantarum P8* 2 × 10^10^ CFU/ day	12	Improvement in anxiety, stress, memory, and cognitive function	PSS-10, DASS-42	The probiotic reduced symptoms of stress and anxiety, accompanied by an improvement in cognitive function and memory.
Wu SI et al., 2022 [84]	I:33 C:32	Anxiety and stress	35.3	98	Capsule; HK-PS23 (300 mg de *L. paracasei* PS23)	8	Improvement in anxiety score	PSS-10, STAI	The probiotic reduced stress and anxiety symptoms through anti-inflammatory properties, followed by an improvement in memory and cognitive abilities.

I/C: Intervention/Control; %f: percentage of women. CFU: Colony Forming Units. BPRS: Brief Psychiatric Rating Scale. BDI-II: Beck Depression Inventory. MADRS: Montgomery–Åsberg Depression Rating Scale. HDRS-24: Hamilton Depression Rating Scale. HADS: Hospital Anxiety and Depression Scale. Ecologic^®^Barrier: consists of 9 strains *Bifidobacterium bifidum W23*, *Bifidobacterium lactis W51*, *Bifidobacterium lactis W52*, *L. acidophilus W37*, *Lactobacillus brevis W63*, *Lactobacillus casei W56*, *Lactobacillus salivarius W24*, *Lactococcus lactis W19*, and *Lactococcus lactis W58* (total cell count 1 × 10^10^ CFU/day). Vivomixx^®^: contains *Streptococcus thermophilus NCIMB 30438*, *Bifidobacterium breve NCIMB 30441*, *Bifidobacterium longum NCIMB 30435* (reclassified as *B. lactis*), *Bifidobacterium infantis NCIMB 30436* (reclassified as *B. lactis*), *Lactobacillus acidophilus NCIMB 30442*, *Lactobacillus plantarum NCIMB 30437*, *Lactobacillus paracasei NCIMB 30439*, *Lactobacillus delbrueckii* subsp. *Bulgaricus NCIMB 30440.* STAI: State-Trait Anxiety Inventory. OMNi-BiOTiC^®^ Stress Repair: *B. bifidum W23*, *B. lactis W51*, *B. lactis W52*, *L. acidophilus W22*, *L. casei W56*, *L. paracasei W20*, *L. plantarum W62*, *L. salivarius W24*, and *L. lactis W19.* HAMA-14: Hamilton Anxiety Scale. SSRIs: Selective Serotonin Reuptake Inhibitors. SCL-90: Derogatis Symptom Checklist. PSS-10: Perceived Stress Scale. DASS-42: Depression, Anxiety, and Stress Scales. NR: not reported. QIDS-SR16: Quick Inventory of Depressive Symptomatology. SAMe: S-adenosylmethionine. Z-SDS: Zung Self-Rating Depression Scale. Multispecies probiotic: *B. subtilis*, *B. bifidum*, *B. breve*, *B. infantis*, *B. longum*, *L. acidophilus*, *L. delbrueckii* subsp. *bulgaricus*, *L. casei*, *L. plantarum*, *L. rhamnosus*, *L. helveticus*, *L. salivarius*, *L. lactis*, and *S. thermophilus.* RBANS: Repeatable Battery for the Assessment of Neuropsychological Status. MPI: Memory Performance Index. VIM: Visual Memory Test. VBM: Verbal Memory Test. VLMT: Verbal Learning Memory Test. ADAS-Jcog: Alzheimer’s Disease Assessment Scale-Cognitive. MMSE: Mini-Mental State Examination. HAM-A: Hamilton Anxiety Rating Scale. *** S. thermophiles strain CNCM number I-1630*, *B. animalis* subsp. *Lacti*, *Bifidobacterium bifidum*, *S. thermophiles*, *L. bulgaricus strain numbers CNCM I-1632 and I-1519*, *L. lactis* subsp. *Lactis strain CNCM number I-1631*, *L. acidophilus*, *L. plantarum*, *L. reuteri*, each strain at 1.5 × 10^10^ CFU. PHQ-9: Patient Health Questionnaire-9. BPD: Bipolar Disorder. PANSS: Positive and Negative Syndrome Scale. BAI: Beck Anxiety Inventory. Sanprobi Barrier^®^: *B. bifidum W23*, *B. lactis W51*, *B. lactis W52*, *L. acidophilus W37*, *L. brevis W63*, *L. casei W56*, *L. salivarius W24*, *L. lactis W19*, and *L. lactis W58*. Bio-Kult^®^: *B. subtilis PXN^®^ 21*, *B. bifidum PXN^®^ 23*, *B. breve PXN^®^ 25*, *B. infantis PXN^®^ 27*, *B. longum PXN^®^ 30*, *L. acidophilus PXN^®^ 35*, *L. delbrueckii ssp.bulgaricus PXN^®^ 39*, *L. casei PXN^®^ 37*, *L. plantarum PXN^®^ 47*, *L. rhamnosus PXN^®^ 54*, *L. helveticus PXN^®^ 45*, *L. salivarius PXN^®^ 57*, *L. lactis ssp.lactis PXN^®^ 63*, *S. thermophilus PXN^®^ 66.* ETB: Emotional Test Battery. LactoCare^®^: *L. acidophilus*, *B. lactis*, *B. bifidum*, *and B. longum* (each 2 × 10^9^). ACPT: Auditory Continuous Performance Test. CES-D: Center for Epidemiologic Studies Depression Scale. Familact H^®^: *L. casaei* 3 × 10^8^ CFU/g, *L. acidophilus* 2 × 10^8^ CFU/g, *L. bulgaricus* 2 × 10^9^ CFU/g, *L. rhamnosus* 3 × 10^8^ CFU/g, *B. breve* 2 × 10^8^ CFU/g, *B. longum* 1 × 10^9^ CFU/g, *S. thermophilus* 3 × 10^8^ CFU/g, and 100 mg of fructooligosaccharide. BioCeuticals^®^: *B. bifidum*, *B. animalis* subsp. *Lactis*, *B. longum*, *L. acidophilus*, *L. helveticus*, *L. casei*, *L. plantarum*, *L. rhamnosus.* GAD-7: Generalized Anxiety Disorder Scale. FamiLact^®^: *L. Acidophilus*, *L. Casei*, *L. Delbrueckii* subsp. *L. Bulgaricus*, and *L. Rhamnosus*, *B. Longum* and *B. Breve* and *S.Salivarius* subsp. *Thermophil*. Probio-Tec^®^ BG-VCap-6.5: *L. rhamnosus* and *B. animalis* subsp. *Lactis.* YMRS: Young Mania Rating Scale.

**Table 2 nutrients-16-01352-t002:** Summary of articles that assessed adverse events in the studied population.

Author, Year	Number of Patients with Psychobiotics	Number of Adverse Events in Patients with Psychobiotics	Reported Adverse Events	Conclusion on Probiotic Safety
Dickerson F et al., 2014 [67]	33	0	Not reported	One serious adverse event occurred in the probiotic group, but none were directly related to the product.
Romijn et al., 2017 [38]	40	63	Constipation, appetite changes, nausea, weight gain, dry mouth, abdominal pain, anxiety, headache, rash, blurred vision, and sleep disruption.	Three serious adverse events occurred during the trial, which were associated with the placebo group. There were no serious adverse events in the probiotic group.
Pinto et al., 2017 [74]	22	4	Rhinitis, constipation.	Of 18 reported adverse events, only 4 were related to the supplied probiotic product.
Rudzki et al., 2018 [77]	30	14	Headache, diarrhea, and flatulence.	No serious adverse events occurred; moreover, patients with gastrointestinal events had irritable bowel syndrome, which could have contributed to these manifestations.
Dickerson F et al., 2018 [70]	26	75	Gastrointestinal, metabolic and endocrine, musculoskeletal, sensory alterations, cardiovascular and respiratory.	The probiotic was well tolerated by participants, and there were no withdrawals from the study related to the product. The authors did not detail the specific type of adverse event and whether these were directly related to probiotic consumption.
Majeed M et al., 2018 [48]	20	0	Not reported.	No adverse events related to probiotic intake occurred during the study period.
Ghorbani Z et al., 2018 [49]	20	3	Nausea and bloating.	The adverse events that occurred were not serious and, therefore, did not lead to any participant withdrawing from the study. There were no differences in the rate of adverse events between groups.
Miyaoka T et al., 2018 [50]	20	3	Neurological and dermatological.	The adverse events presented by participants were not detailed. No serious adverse events related to the probiotic were reported.
Kazemi A et al., 2018 [51]	74	13	Gastrointestinal issues, fever, body pain, and increased appetite.	No serious adverse events related to the consumption of probiotics or prebiotics were reported.
Vaghef E et al., 2021 [46]	22	5	Gastrointestinal discomfort.	No serious adverse events occurred. The adverse events that occurred were resolved in less than two weeks.
Kobayashi Y et al., 2019 [62]	61	0	Not reported.	No adverse events related to probiotic intake occurred during the study period.
Hwang Y et al., 2019 [61]	50	7	Dizziness, stomach aches, headaches, gastritis, erectile dysfunction, and seborrheic dermatitis.	One of the adverse events reported in the probiotic group was classified as serious, and the participant withdrew to receive treatment. Most adverse events were classified as mild.
Gualtieri et al., 2020 [72]	65	0	Not reported.	No adverse events were observed in the study.
Xiao et al., 2020 [58]	40	0	Not reported.	No adverse events were observed in the study.
Sacarello et al., 2020 [39]	45	2	Reduced appetite and mood disorder.	The reported adverse events were not related to the product supplied in the study.
Zhang et al., 2021 [41]	38	0	Not reported.	No adverse events were observed in the study.
Eskandarzadeh S et al., 2021 [73]	24	2	Dizziness and itching.	No serious adverse events leading to participant withdrawal were reported.
Haghighat N et al., 2021 [79]	25	1	Headache.	No serious adverse events occurred, and the one that occurred in the probiotic group was not directly related to the administration of the product.
Regiada L et al., 2021 [80]	27	7	Burping, bloating, increased frequency of bathroom visits, acne, anxiety, and insomnia.	No serious adverse events occurred. It is not specified whether the events were directly related to the intake of psychobiotics.
Jamilian H et al., 2021 [64]	26	0	Not reported.	No adverse events were observed in the study.
Asaoka et al., 2022 [60]	55	1	Constipation.	Three adverse events (including constipation) occurred; however, it is concluded that they were not related to probiotic consumption.
Vaghef E et al., 2023 [47]	22	5	Flatulence and soft stools.	No serious adverse events leading to participant withdrawal were reported.
Nikolova et al., 2023 [40]	24	14	Nausea, diarrhea, indigestion, constipation, reflux, heartburn, stomach pain, and burping.	There were no serious adverse events, nor treatment discontinuation associated with this cause. Nausea and indigestion only occurred in the probiotic group.
Freijy T et al.,2023 [81]	63	22	Bloating, gas, abdominal discomfort, changes in bowel movements, and headaches.	The treatment was well tolerated and caused few adverse events. No adverse events led to withdrawal from the study.

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
