# Peer review of "Effectiveness of Psychobiotics in the Treatment of Psychiatric and Cognitive Disorders: A Systematic Review of Randomized Clinical Trials"

_nutrients, 2024, doi:10.3390/nu16091352_

Round 1
Reviewer 1 Report
Comments and Suggestions for Authors
1 - Could the authors provide more information on these non-retrieved papers?
2 - Please report the details surrounding inter-rater reliability.
3 - There is substantial heterogeneity in psychiatric disorders (the ones included in the studies), and the traditional treatment strategies for these psychiatric disorders span a wide range. Discussion of this is needed.
4 – I would not consider “stress” a DSM disorder. This either needs to be clarified (e.g., trauma and stressor-related disorders) or might consider excluding from the systematic review.
5 – The discussion would benefit from specific mentioning of effect sizes reported in these studies. It is worth commenting on the magnitude of the effects, as they provide important context for understanding this emerging literature.
6 – At its present form, it is unclear the review and future directions that the authors suggest. The paper would be strengthened by going beyond just summarizing prior literature. For example, it would be helpful to include a discussion of gaps in the current literature, hypotheses, and suggested next steps. Additionally, a section focused on limitations would be helpful, in view of the widely inconsistent effects reported (and very few replication studies, and studies cannot be compared across due to differences in, for example, how psychiatric disordered was diagnosed (self-report; clinical interview), treatment duration, etc).
7 – The paper would benefit from a careful proofreading throughout and editing for clarity.
8 – Abstract: please clarify what this means: “showing a high assessment in depression”
Comments on the Quality of English LanguageThe paper would benefit from a careful proofreading throughout and editing for clarity.
Author Response
We deeply appreciate your detailed and constructive observations. Below, we present a summary of the changes made to our manuscript in response to your comments. We hope the changes made have addressed the concerns raised and improved the quality and clarity of the manuscript.
1 - Could the authors provide more information on these non-retrieved papers?
Response:
During the study selection phase for our systematic review, we identified a total of 5360 potentially relevant articles through our searches in established databases. After removing 239 texts due to duplications and examining 4449 articles by title, we eliminated an additional 583 after reviewing titles and abstracts. Of the 89 studies investigated further, 37 were excluded for various specific reasons, detailed below:
- Four studies were not randomized controlled trials, which was a necessary inclusion criterion to ensure the quality and methodological consistency required for our review.
- Two studies involved underage patient populations, which were excluded because our focus was on adults.
- Twenty-four studies, although measuring changes in intestinal microbiota or biomarkers, did not evaluate the outcomes previously established in our review, meaning they did not directly address the effects of psychobiotics on the psychiatric and cognitive disorders of interest.
- One study was not available in an accessible format, preventing its detailed analysis.
- Four studies examined pregnant women, who were excluded from our study due to the potential confounding variables related to hormonal and physiological changes during pregnancy that could affect the intestinal microbiota.
- Two studies evaluated patients with neurodegenerative or structural diseases such as Huntington's or Parkinson's, which were excluded because our focus was on psychiatric and cognitive disorders without these neurodegenerative components.
2 - Please report the details surrounding inter-rater reliability.
Response: It was added in the results.
3 - There is substantial heterogeneity in psychiatric disorders (the ones included in the studies), and the traditional treatment strategies for these psychiatric disorders span a wide range. Discussion of this is needed.
Response: It was added in the discussion section.
4 – I would not consider “stress” a DSM disorder. This either needs to be clarified (e.g., trauma and stressor-related disorders) or might consider excluding from the systematic review.
Response: It was removed because it was not part of a DSM disorder.
5 – The discussion would benefit from specific mentioning of effect sizes reported in these studies. It is worth commenting on the magnitude of the effects, as they provide important context for understanding this emerging literature.
Response: It was added in the discussion section.
6 – At its present form, it is unclear the review and future directions that the authors suggest. The paper would be strengthened by going beyond just summarizing prior literature. For example, it would be helpful to include a discussion of gaps in the current literature, hypotheses, and suggested next steps. Additionally, a section focused on limitations would be helpful, in view of the widely inconsistent effects reported (and very few replication studies, and studies cannot be compared across due to differences in, for example, how psychiatric disordered was diagnosed (self-report; clinical interview), treatment duration, etc).
Response: It was added in the discussion section.
7 – The paper would benefit from a careful proofreading throughout and editing for clarity.
Response: The discussion section was redesigned for greater clarity.
8 – Abstract: please clarify what this means: “showing a high assessment in depression”
Response: Abstract was modified to clarify this statement.

Reviewer 2 Report
Comments and Suggestions for Authors
I appreciate the topic very much. Your survey is excellent (maybe your vision into the future concerning AL is too optimistic).
I have only one remark , suggestion.
Nowadays many probiotics are commercially produced and easily available, with different content of active substances , recommended for all ages from babies to old people, mostly for preventive reasons. Coud you briefly comment on that ? What is your opinion ? Is it with current knowledge acceptable?
Usage of new technologies is a nice vision, however, in common clinical practice where we are?
Author Response
Dear,
We deeply appreciate your comments and observations, which undoubtedly enrich our work. Your point regarding the possible over-optimism about the future of AI is well taken, and we will consider a more balanced presentation of potential future developments.
We recognize that a wide variety of probiotics are now commercially available, with formulations that significantly vary in terms of the content of active substances. These products are accessible to consumers of all ages and are primarily promoted for preventive purposes.
Based on current scientific literature, we consider the use of probiotics to generally be safe and potentially beneficial for health, particularly in preventing gastrointestinal disorders and strengthening the gut microbiota. Strains such as Lactobacillus and Bifidobacterium, as detailed in our systematic review, are often used in treatments lasting from 4 to 24 weeks, depending on the individual's needs and the specific health outcomes desired.
Despite their widespread non-prescription availability, the integration of probiotics, and particularly psychobiotics, into standard clinical practice presents challenges. However, it is crucial that consumers select probiotic products that have proven efficacy and safety in rigorous clinical studies. The efficacy of a probiotic can vary widely depending on the strain or combination of strains used, as well as the dosage and quality of the product. This underscores the variability in treatment outcomes and the critical need for personalized treatment plans based on individual health conditions and probiotic strains.
The use of advanced technologies, such as artificial intelligence, in clinical practice is still in its early phases, especially in areas outside of major urban medical centers. Although these technologies offer promising possibilities for improving medical care, their practical implementation must be carefully managed to ensure that health outcomes are improved without increasing existing disparities in access to healthcare.
Thank you once again for your inquiries, and we hope as the authors of the review to have provided the answers you were looking for.